# Fabrication and Characterization of Mineral Hydrophilic Antifogging Film via Vacuum Evaporation Method

**Shenyu Wei** [1,2,3,4], **Qi Zheng** [1,2,3,4,*], **Lei Wang** [1,2,4,*], **Cheng Peng** [1,2,3,4], **Xinglan Cui** [1,2,3,4], **Xiaokui Che** [1], **Wuyu Wang** [2] and **Zeen Yu** [1,2,3,4]

[1]  National Engineering Reasearch Center for Environment-Friendly Metallurgy in Producing Premium Nonferrous Metals, China GRINM Group Co., Ltd., Beijing 100088, China
[2]  GRINM Resources and Environment Tech. Co. Ltd., Beijing 100088, China
[3]  General Research Institute for Nonferrous Metals, Beijing 100088, China
[4]  Beijing Engineering Research Center of Strategic Nonferrous Metals Green Manufacturing Technology, Beijing 100088, China
*  Correspondence: mr1311@grinm.com (Q.Z.); wanglei@grinm.com (L.W.); Tel.: +86-10-13910161083 (Q.Z.)

**Abstract:** Natural silicate minerals have a wide range of applications as green, non-toxic and low-cost materials. In this paper, hydrophilic anti-fog films of silicate minerals were generated via a vacuum evaporation coating method using natural feldspar minerals as raw material. Thermogravimetric analysis shows that the feldspar changes its structure during the coating process, which in turn changes the roughness of the film and improves the hydrophilicity of the film. The hydrophilicity, anti-fogging properties, optical properties and surface morphology of the films were characterized by contact angle measurements, the hydrothermal method, UV-VIS spectrophotometer and atomic force microscopy, respectively. The results show that the mineral films have excellent hydrophilicity. The best anti-fog effect was achieved at a minimum contact angle of 22.3° with water when the thickness of the film was 100 nm. The anti-fog effect gradually decreased with the increasing thickness of the film. The optical transmittance test showed that the film material had a negligible effect on the transmittance of the substrate. When the film thickness was 100 nm, the maximum optical transmittance was 92.2%. This is 4.5% higher than when the film was uncoated, which has a specific visual transmittance effect.

**Keywords:** hydrophilic films; vacuum evaporation coating; anti-fogging; silicate minerals

## 1. Introduction

Various transparent material products, such as glass windows, agricultural canopies and goggles, are widely used because of their superior light transmission. However, in practice, the surface of materials always fogs up due to temperature changes in the environment, which significantly affects the regular use of the material. In today's society, the anti-fog issue is relevant not only to medical care or people's livelihoods [1–3] but also to aerospace [4–13] and sports [14].

At present, there are two main ideas to prevent the fogging of transparent material surfaces. One is to change the environmental parameters of the substrate, that is, to heat the transparent substrate or increase the airflow rate on the surface of the substrate. This can effectively make the transparent material have anti-fog properties. However, this idea requires specific equipment and consumes energy, which significantly limits the application of this method. Another is to change the surface characteristics of the transparent substrate, such as the chemical composition and roughness of the substrate surface, or to adjust the hydrophobicity of the substrate surface by depositing functional coatings [15,16]. When a water droplet comes into contact with a superhydrophobic surface, it retains its original form and requires an inclined plane to keep the surface dry. When the droplet comes into contact with a superhydrophilic surface, the droplet condenses on the superhydrophilic

surface as a thin film. Compared with a superhydrophobic surface, the thin film droplets can not only reduce the reflection and scattering in the water droplets, but also have a faster evaporation rate. A superhydrophilic surface can be dried in a short time [17]. This unique advantage led superhydrophilic films to gradually become the research hotspot of anti-fog materials.

The main methods used to prepare hydrophilic anti-fog films are sol-gel, physical vapor deposition (PVD) and electrochemical deposition. The sol-gel method [18,19] is when metal salts undergo hydrolysis or condensation reactions in a suitable solvent to form a gel. The gel is coated on the surface of the substrate and then dried to obtain a hydrophilic coating. This method requires low substrate requirements and high product purity. For example, Chen et al. [20] used the sol-gel method to prepare ZnO sol-gel and $TiO_2$ sol-gel. They used the spin-coating method to prepare $ZnO/TiO_2$ composite coatings on glass substrates, which can achieve super-hydrophilic properties under light-free conditions. Still, it has the disadvantages of a relatively long preparation process and organic solvent pollution. Physical vapor deposition (PVD) [21] is a method of physically obtaining a gaseous coating material under vacuum and then depositing it on the substrate surface to form a hydrophilic coating. For example, Rico et al. [22] deposited a 300–500 nm thick $TiO_2$ layer on the surface of silicon wafers by PVD at 400 °C. The coating has a porous structure and high roughness and can reach a superhydrophilic state under UV light irradiation, but its preparation cost is high. The electrochemical deposition method [23–25] is used to obtain materials with different morphologies and microscopic dimensions by adjusting parameters such as current density, deposition time, and temperature. It can adjust the superhydrophilicity of the material from structural and morphological directions. You et al. [26] used the electrodeposition method to achieve the preparation of Zn/ZnO crystals by electrodeposition on a copper grid. The electrodeposition process was optimized by varying the applied voltage and duration. The all-inorganic film exhibited superhydrophilicity in air and could be used as an efficient oil–water separation device. In summary, the current preparation of anti-fog materials relies on organic substances or metal salts/metal oxides to prepare hydrophilic anti-fog films. Most of these have disadvantages such as being harmful to humans, polluting the environment or being expensive. By contrast, some natural non-metallic minerals are safe, environmentally friendly and inexpensive, and some non-metallic minerals such as mica, feldspar, quartz and other silicate minerals have good natural hydrophilic properties. Their prepared film materials may have potential anti-fogging properties.

In this study, a new mineral hydrophilic antifogging film was prepared using natural feldspar minerals as raw materials. The coating process was optimized to obtain a green, non-toxic significant antifogging coating. The effect of film thickness on the above properties was investigated through thermogravimetric analysis of the coating materials, characterization of the anti-fog properties, hydrophilicity, surface morphology, roughness and optical transmittance of the films with different layer thicknesses, and the relationship between roughness and hydrophilic anti-fog properties was highlighted.

## 2. Materials and Methods

### 2.1. Materials

In this experiment, natural feldspar minerals were selected as the coating materials, all of which are of industrial quality. The multielement chemical analysis results of feldspar minerals is shown in Table 1.

**Table 1.** Multielement chemical analysis results of feldspar minerals.

| Element | $Al_2O_3$ | $SiO_2$ | $K_2O + Na_2O$ | $CaO + MgO$ | $Fe_2O_3$ | $TiO_2$ |
|---|---|---|---|---|---|---|
| Cotent % | 19.50 | 65.70 | 13.60 | 0.30 | 0.05 | 0.04 |

The mineral hydrophilic anti-fog film was prepared as follows: (1) Experimental preparation: preparation of the coating raw materials required for the experiment. The

mineral powder was mixed with 5 wt% distilled water and put into a tabletting machine (YP-15, JOSVOK, Tianjin, China) at 40 Mpa for 10 min. We then put the mineral billet into a Muffle furnace (TMX-12-18, FNS, Beijing, China) for 2 h at 500 °C, took it out and cooled to obtain the coating material. The polycarbonate (PC) substrate required for the experiment was ultrasonically cleaned with ethanol anhydrous (purity ≥ 99.5 wt%, Macklin, Shanghai, China) and deionized water. The single ultrasonic cleaning time had to reach 10 to 15 min, so that the oil and impurities on the surface of the substrate could be effectively removed. (2) Instrument preparation: The condensate circulation was opened in advance before the vacuum coating mechanism (ZZS-900, Chengdu Vacuum Machinery Factory, Chengdu, China) was turned on. After opening the vacuum coating room, we cleaned the coating room and kept it clean. We then installed the required coating material. After confirmation, we closed the coating room. (3) When the vacuum degree in the coating machine was $3 \times 10^{-3}$–$9 \times 10^{-3}$ Pa, the coating operation began. The substrate temperature was 60 °C, and the coating rate was controlled to be 3–7 Å/s. The physical vapor deposition was used to vaporize the pressed raw material onto the polycarbonate (PC) base at high temperature, and the mineral material hydrophilic anti-fog film was obtained. Then, the experimental operation was carried out according to the corresponding experimental parameters. After the entire coating process was completed, the sample was taken out when the temperature in the coating room was lower than 50 °C. After removal, we closed the coating room and kept the coating room. Finally, we turned off the cold circulating water. Figure 1 shows a schematic diagram of the mineral anti-fog film material preparation.

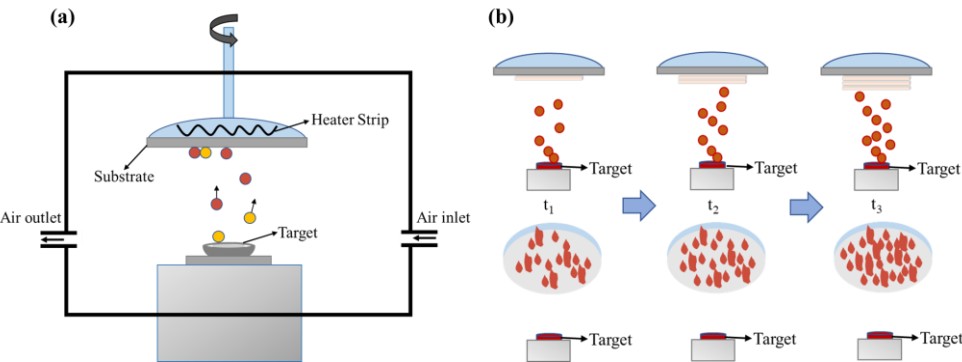

**Figure 1.** The diagram of preparing mineral anti-fog film materials. (**a**) Device diagram of vacuum evaporation coating machine; (**b**) growth of films with different thicknesses.

*2.2. Characterization*

The TG-DTA8122 thermogravimetric analyzer (Rigaku, Kyoto, Japan) was used for differential thermogravimetric analysis of raw minerals. The test atmosphere was nitrogen atmosphere, the reference was blank, the heating rate was 10 °C/min, and the scanning temperature range was 30–1000 °C. The contact angle of the film surface to deionized water was measured using a DSA100s contact angle tester (Kruss, Hamburg, Germany), which determined the hydrophilicity of the mineral hydrophilic anti-fog film. Deionized water was added to the syringe, and the drop amount was set to 4 µL/drop. Five different points on the coating were selected to test the contact angle, and the average value was calculated as the final contact angle of the coating. The surface morphology and surface roughness of the mineral hydrophilic antifogging film were observed using a Dimension Icon atomic force microscope (Bruker, Karlsruhe, Germany). In the test, the scanning mode was tap mode, the silicon probe was used, the elastic coefficient was 1.7 N/m, and the scanning frequency was 1 Hz. The optical transmittance of the mineral hydrophilic antifogging film and substrate was measured using a UV-2600 ultraviolet-visible spectrophotometer (Shimadzu, Kyoto, Japan). The background was air, and the scanning wavelength was 200–1200 nm.

### 2.3. Anti-Fogging Performance Test

The antifogging performance of the mineral hydrophilic antifogging film was tested using the rapid thermal fogging method (GB/T 31726-2015). It operates by filling a beaker with 200 ± 10 mL of distilled water at 85 ± 2 °C and quickly clamping the antifogging performance test face of the sample to the mouth of the beaker. At that point, the timing lasted for 60 s. After 60 s, the surface of the film sample was observed under natural light. The sample was observed within 5 s. Finally, we recorded the anti-fog rating of the sample.

## 3. Results

### 3.1. Thermogravimetric Analysis of Coating Materials

The thermogravimetric analysis of the coated raw material shown in Figure 2 indicates that the weight loss of the sample was under 400 °C. From 400 °C, the thermogravimetric curve of the sample drops sharply. There is a noticeable peak of heat absorption in the temperature range of 400–600 °C. The weight loss of the sample at this time is mainly due to the detachment of interlayer water and bonded water, but the crystal structure does not change. The weight of the mineral continues to drop at 600–850 °C. At this stage, the combined and interlayer water is gradually detached from the sample while the crystal structure of the mineral is maintained. At 850–950 °C, there is another central valley of heat absorption. At this point the detachment of the combined and interlayer water ends and the crystal structure of the mineral is altered. The sample mass loss measured over the entire test temperature was 4.30%. Through the thermogravimetric analysis of the coating material, it is speculated that the structure of the coating material will change in the vacuum evaporation coating. This provides a reference for evaporation coating.

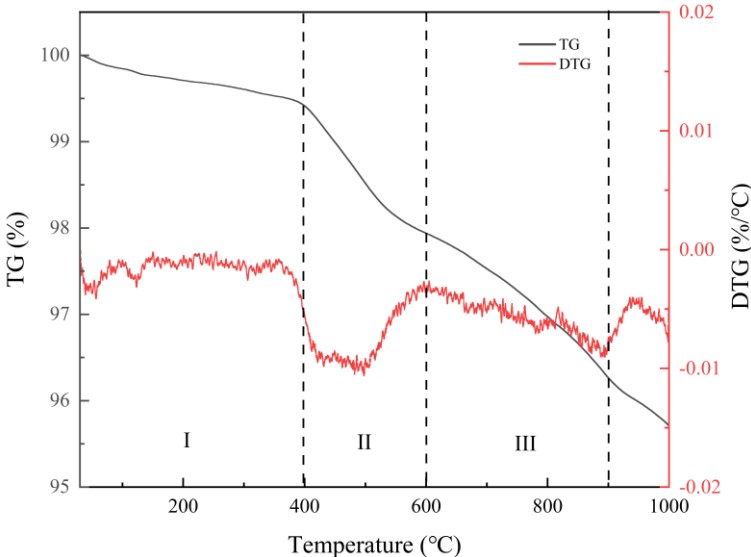

**Figure 2.** DTG analysis diagram of coating raw materials.

### 3.2. Analysis of Film Thickness and Anti-Fog Performance

In the vacuum coating process, as the coating time increases, the deposition thickness of the mineral hydrophilic anti-fog film increases, and thus the morphology of the sample surface changes. Thus, the film thickness has a specific influence on the anti-fog effect of the sample. In the experiment, the influence of film thickness on the anti-fog effect of film was investigated. The thickness of the film layer was set as 100, 200, 400, 800 and 1000 nm. We then placed the film samples above the 85 ± 2 °C distilled water. The anti-fog effect of the samples was observed at room temperature.

Figure 3 shows the anti-fogging effect of film samples with different film thicknesses. The anti-fogging effect can be classified into different grades by the national standard of the rapid thermal fogging method (GB/T 31726-2015). Table 2 shows the anti-fogging

grade table for film samples with different film thicknesses. As seen from Figure 3 and Table 2, the anti-fog effect of samples with a film layer thickness between 100–200 nm was superior, with an anti-fog grade of 1. The surface of sample is transparent and clean without water droplets. The lettering under the sample can be seen clearly. When the thickness of the film reached 400 nm or more, water droplets appeared on the surface of the film samples, which dramatically affects their transparency. In Figure 3, the writing under the substrate is blurred, and the anti-fog effect is poor. When the film thickness is 1000 nm, the film samples have a thickness, which makes the anti-fog effect even worse, and the anti-fog grade only reaches 4. From the above analysis, the hydrophilic and antifogging film formed on a PC substrate by vacuum evaporation coating of natural mineral materials has a certain antifogging effect. Additionally, when the thickness of the film is 100–200 nm, the antifogging effect is the best.

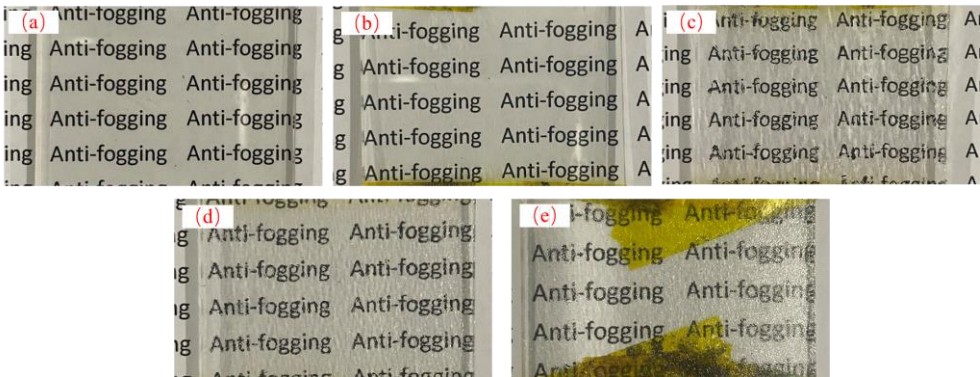

**Figure 3.** Anti-fog effect of film samples with different film thicknesses: (**a**) 100 nm; (**b**) 200 nm; (**c**) 400 nm; (**d**) 800 nm; (**e**) 1000 nm.

**Table 2.** Anti-fog rating table of film samples with different film thicknesses.

| Thickness (nm) | 100 | 200 | 400 | 800 | 1000 |
|---|---|---|---|---|---|
| Anti-fog rating | Grade 1 | Grade 1 | Grade 2 | Grade 3 | Grade 4 |

### 3.3. Film Thickness and Hydrophilicity Analysis

In order to study the influence of film thickness on the surface hydrophilicity of mineral hydrophilic antifogging film, the contact angles of films with different thicknesses were measured.

Figure 4 shows the contact angle analysis on the surface of the film samples with different film thicknesses. As seen in Figure 4, when the uncoated transparent substrate surface was in contact with the water droplets, the droplets always condensed into a mass and did not spread out on the substrate surface. The surface substrate was less hydrophilic. The contact angle of the sample was 75.1°, as seen in Table 3. When mineral materials are plated on the surface of the substrate, water droplets spread on the surface of the film sample. The hydrophilicity is obviously improved compared with the blank substrate. With the increase of the thickness of the film, the contact angle of the film sample gradually increases and the hydrophilicity gradually decreases. This phenomenon also corresponds to the anti-fog grade of the film sample in Table 1. According to Young's equation [27], when the contact angle is less than 90°, the surface of the film presents hydrophilic properties. The smaller the contact angle, the stronger the surface hydrophilic properties. It can be seen from Table 2 that the final contact angle of the film sample with a thickness of 100 nm is 22.3° after 5 s. At this time, the film sample exhibits the best hydrophilicity and water droplets spread evenly on the surface of the film. This not only helps reduce the scattering and reflection at the water droplets and enhance the anti-fog performance of the film, but also speeds up the evaporation of water droplets on the film, thus making the film drier.

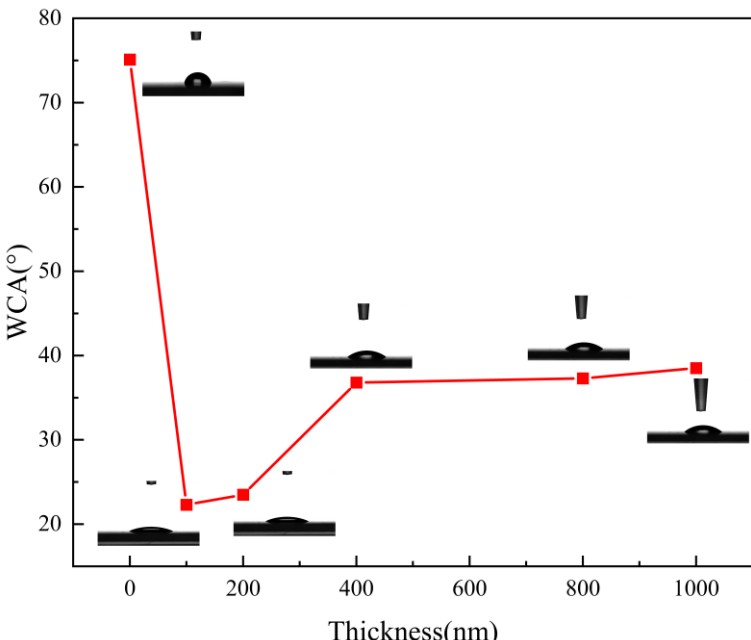

**Figure 4.** Surface contact angles of film samples with different film thicknesses.

**Table 3.** Contact angles of film samples with different film thicknesses.

| Thickness (nm) | 0 | 100 | 200 | 400 | 800 | 1000 |
|---|---|---|---|---|---|---|
| Contact angle (°) | 75.1 | 22.3 | 23.5 | 36.8 | 37.3 | 38.5 |

### 3.4. Film Thickness and Surface Morphology Analysis

AFM was used to characterize the surface structure and roughness of the silicate mineral hydrophilic anti-fog films. Figure 5 shows the AFM plots of the film samples with different thicknesses. It can be seen from Figure 5 that the surface of the film samples has a nano-scale microstructure. The difference in density of the nano-emulsion structure leads to a change in the surface roughness of the samples. Combining the AFM plots of the film samples in Figure 5 with the surface roughness analysis of the films in Table 4, it can be seen that when the thickness of the film samples is in the range of 100–200 nm, the surface roughness of the samples is high, corresponding to an anti-fog effect of up to class 1. When the film thickness reaches 400 nm or more, the surface roughness of the film sample decreases, and the anti-fog effect gradually diminishes. From the Wenzel equation [28], it is clear that increasing roughness enhances the wettability of the sample surface, resulting in a more hydrophilic substrate. This in turn gives the substrate its anti-fog properties. During the coating experiments, the mineral particles evaporated at high temperatures to form nano-sized mineral particles, randomly adhering to the substrate surface. This resulted in a relatively higher roughness and increased hydrophilicity relative to the original substrate surface. With the extension of the vacuum coating time, the thickness of the film layer gradually increases. The corresponding nano-mineral particles on the substrate surface increase in density, reduce the accumulation gap, and reduce the roughness, making the film hydrophilicity appear to decline. Therefore, the nano-mineral particles deposited on the substrate increase the roughness of the substrate surface, making it macroscopically show hydrophilic properties and play an anti-fog role. This is the main reason for the good anti-fog performance of the mineral hydrophilic anti-fog film.

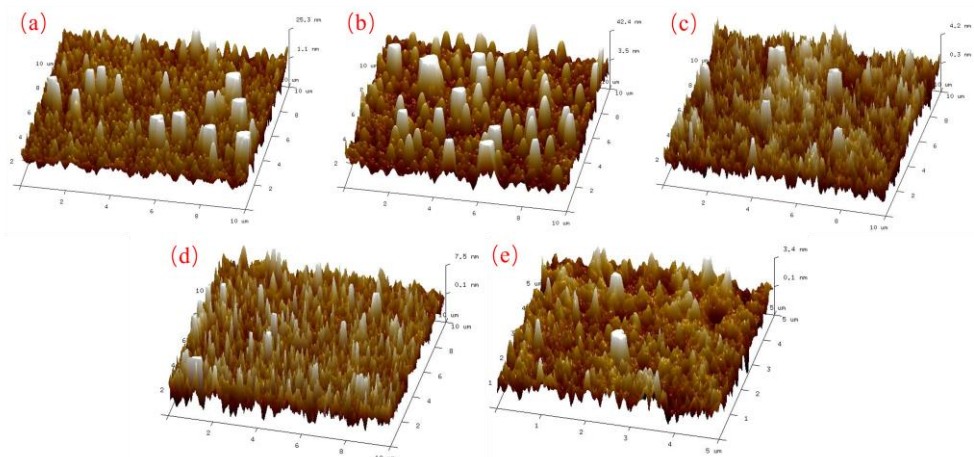

**Figure 5.** AFM images of thin film samples with different film thicknesses: (**a**) 100 nm; (**b**) 200 nm; (**c**) 400 nm; (**d**) 800 nm; (**e**) 1000 nm.

**Table 4.** Surface roughness of thin film samples with different thicknesses.

| Thickness (nm) | 100 | 200 | 400 | 800 | 1000 |
|---|---|---|---|---|---|
| Ra (nm) | 26.195 | 28.958 | 4.203 | 5.603 | 4.348 |

From the above analysis, it can be seen that the substrate surface roughness and material anti-fog performance are closely related. According to the Wenzel equation, a liquid in contact with a solid's rough surface completely penetrates the groove, which makes the contact area between the solid and liquid increase. We thus introduce the roughness factor r, and define r as the ratio of the actual surface area of the solid surface to the apparent surface area, and its theoretical equation is:

$$\cos\theta_W = r\cos\theta \tag{1}$$

where $\theta_W$ is the droplet on a rough surface, and $\theta$ is the contact angle of the droplet on a smooth surface. Usually $r \geq 1$. From the equation, it can be found that as roughness increases, hydrophilic surfaces ($\theta < 90°$) become more hydrophilic, indicating that an increase in surface roughness further amplifies the wetting performance of smooth materials [29]. Therefore, the surface contact angle of a material depends on its intrinsic contact angle and the surface roughness of the material. The intrinsic contact angle of a material is the contact angle on a perfectly smooth surface. It can be increased or decreased by changing the material's surface energy using physical or chemical modifications. The hydrophilic properties of the material surface can be achieved by increasing the surface energy and roughness of the material. This leads to calculations of how nanoparticles are deposited on the substrate to further reveal the relationship between surface structure, roughness and surface hydrophilic and anti-fogging properties.

In preparing hydrophilic mineral films by vacuum evaporation, how the mineral particles accumulate on the substrate surface influences the roughness of the film and, thus, its hydrophilicity. During the evaporation of mineral particles, the size, shape and distribution of the particles influence the effect of the film deposition. As mineral particles of different shapes and dimensions are randomly deposited on the surface of the substrate during the coating process when the coating time is short, i.e., when the deposited film is thin, mineral particles of various sizes and shapes exist on the surface of the film, and the surface roughness is high; while as the coating time increases, that is, when the deposited film becomes thicker, mineral particles gradually accumulate on the surface of the substrate, filling the gaps between particles of different sizes and shapes, so that the film's surface tends to be smoother, and the surface roughness is reduced. Thus, different sizes and numbers of particles increase the surface roughness of the film and make it

hydrophilic. Assuming that the mineral particles are spherical and uniform in size, ideally, there are two elemental particle distributions: cubic pile-up and hexagonal densest pile-up. Figure 6 shows the arrangement patterns of the two mineral particles at the beginning of the deposition.

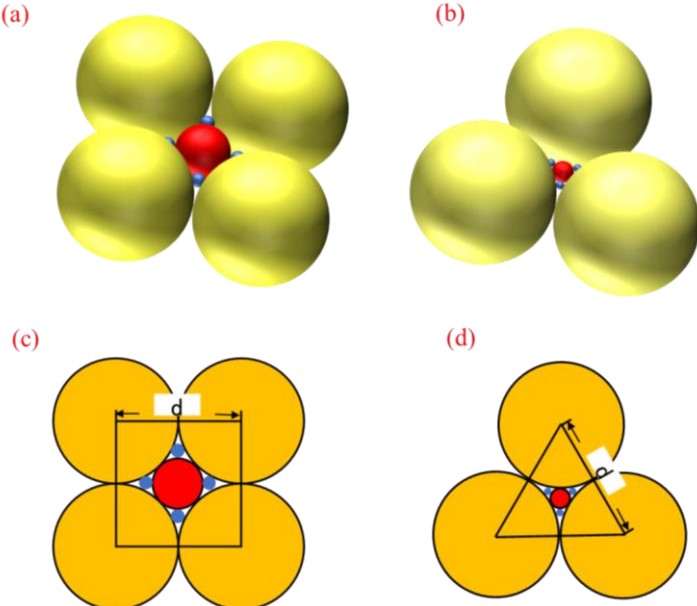

**Figure 6.** Two basic patterns of particle distribution: (**a**) cubic stacking stereogram; (**b**) hexagonal densest stacking stereogram; (**c**) cubic stacking plane graph; (**d**) hexagonal densest stacking plane graph.

To facilitate the calculation of the surface roughness of the material, the surface roughness factor of the material is calculated when the small spheres are overhanging the center of the void created by the stack of large spheres, assuming the uppermost layer of the particle stack as the reference surface and considering only a single layer of particle arrangement. In the case of cubic stacking, the centers of the four adjacent spheres are connected to obtain a square with side length d. Taking the area of the square as the unit area, the actual location of the material in the unit area consists of four 1/8 large spherical spheres, one 1/2 medium spherical sphere and four 1/2 small spherical spheres. The surface roughness factor of the material is calculated as:

$$r = S_r/Sb = \left[\frac{\pi d^2}{8} \times 4 + \frac{(0.414d)^2\pi}{2} + \frac{(0.108d)^2\pi}{2} \times 4\right]/d^2 = 0.61\pi \approx 1.92 \quad (2)$$

In the case of hexagonal closest stacking, a positive triangle with side length d is obtained by connecting the centers of three adjacent spheres, and the triangle is noted as the unit area; then, the actual area of the material surface per unit area consists of three 1/12 large sphere surfaces, one 1/2 medium sphere surface and three 1/2 small sphere surfaces, and the surface roughness factor of the material is calculated as:

$$R = S_r/Sb = \left[\frac{\pi d^2}{12} \times 3 + \frac{(0.156d)^2\pi}{2} + \frac{(0.062d)^2\pi}{2} \times 3\right]/\frac{\sqrt{3}d^2}{4} = 0.62\pi \approx 1.95 \quad (3)$$

The theoretical values of the roughness factor formed by the cubic packing and the hexagonal closest packing model are discussed above. It can be seen from the results that the roughness factor is between 1.92 and 1.95 when the particles of different sizes are stacked. Similarly, we calculated the particle packing model with homogeneous size, whose roughness factor is between 1.51 and 1.81. Therefore, when particles of different

sizes accumulate, the roughness factor is larger, and the surface of the material is rougher. Combined with the actual coating process, from the AFM results of Figure 5, it can be seen that when the film thickness is 100–200 nm, the sample surface has fluctuating mineral particles of different sizes, and the roughness of the film is relatively large. As the film layer gradually thickened to 800–1000 nm, the size and shape of the mineral particles deposited on the surface of the sample gradually became uniform, and the roughness of the film was small. Therefore, when the particles that make up the film are of different sizes, the surface of the material is rougher, so that the surface hydrophilicity is higher, which shows an excellent anti-fog effect.

### 3.5. Film Thickness and Optical Transmittance Analysis

The films were tested for optical transmittance using a UV-VIS spectrophotometer to characterize the optical properties of mineral hydrophilic anti-fog films with different layer thicknesses. Figure 7 shows the optical transmittance of film samples with different layer thicknesses. It can be seen from Figure 7 that in the visible region of 400–760 nm wavelengths, the film sample with a film layer thickness of 100 nm had the highest transmittance of up to 92.2%. This is 4.5% higher than that of the blank substrate. The film samples with a film layer thickness of 200 and 400 nm had a similar transmittance of about 88%. In the wavelength range of 530–660 nm, the transmittance was about 3% higher than that of the blank substrate. This indicates that the mineral hydrophilic film does not affect the light transmission performance of the original substrate material, and the film layer has a specific optical permeability effect on the original substrate material. When the thickness of the film layer reached 800 and 1000 nm, the optical transmittance of the film samples fluctuated wildly. During the coating process, the silicate mineral material was randomly deposited on the substrate. Optical transmittance is influenced by the thickness and uniformity of the nano-mineral particles deposited on the surface of the substrate, which leads to decrease of the optical transmittance of the film. In the infrared range above 760 nm, the best optical transmittance was achieved for the 100 nm layer thickness, followed by the 200 and 400 nm thicknesses, while the worst visual performance was found for the 800 and 1000 nm thicknesses.

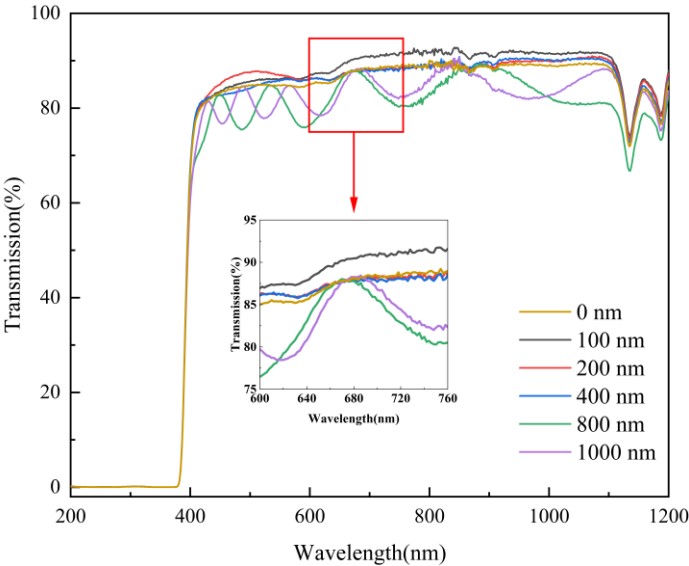

**Figure 7.** Transmittance of film samples with different film thicknesses.

In summary, the optical transmittance of the film samples with different layer thicknesses shows that the mineral hydrophilic anti-fog film with a layer thickness of 100 nm had the best optical properties. Up to 400 nm, the optical transmittance of the original substrate was not affected by the layer thickness and the visual transmittance increased to

a certain extent. As the layer thickness increased, the optical properties of the film samples gradually decreased.

## 4. Conclusions

In this study, mineral hydrophilic anti-fog films were successfully prepared by physical vacuum evaporation coating using natural feldspar minerals as raw materials. The thermogravimetric analysis of the coating material showed that during the coating experiment, the structure of the mineral material changed and the mineral particles were gradually deposited on the substrate surface, thus forming mineral hydrophilic anti-fog films. The anti-fog properties, hydrophilicity, surface morphology, roughness and optical transmittance of different film thicknesses were compared and analyzed. We found that the mineral hydrophilic anti-fog film had a better anti-fog effect when the layer thickness was 100 nm. The minimum contact angle was 22.3° with water and the large surface roughness was 26.195. The film had excellent hydrophilic, anti-fog and optical properties. However, with the increasing thickness of the film layer, the mineral nanoparticles accumulated more and more densely on the surface. The roughness of the formed film decreased gradually, which made the hydrophilicity of the film decrease gradually, and the anti-fog effect and optical properties of the film decrease gradually.

The results of this study show that the films prepared from raw mineral materials have a remarkable anti-fogging effect, stable performance and green environmental protection. The raw materials used are widely available. This study not only opens up the application scenario of non-metallic mineral raw materials but also provides a practical basis for the preparation of new mineral anti-fogging materials. This is of great significance in the research of new materials.

**Author Contributions:** Conceptualization, S.W., Q.Z. and L.W.; methodology, S.W., Q.Z., C.P. and L.W.; software, S.W.; validation, S.W., Q.Z. and L.W.; formal analysis, S.W.; investigation, S.W., Q.Z., L.W. and Z.Y.; resources, S.W., Q.Z. and L.W.; data curation, S.W.; writing—original draft preparation, S.W.; writing—review and editing, Q.Z., L.W. and W.W.; supervision, Q.Z. and L.W.; project administration, Q.Z., L.W., X.C. (Xinglan Cui) and X.C. (Xiaokui Che). All authors have read and agreed to the published version of the manuscript.

**Funding:** National Key Research and Development Program of China 2021YFC2900900, and the Research on mineral anti-fog film material for goggles.

**Institutional Review Board Statement:** Not applicable.

**Informed Consent Statement:** Not applicable.

**Data Availability Statement:** Not applicable.

**Acknowledgments:** The authors gratefully acknowledge the financial support from the following project: National Key Research and Development Program of China 2021YFC2900900, and the Research on mineral anti-fog film material for goggles.

**Conflicts of Interest:** The authors declare no conflict of interest.

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
