# Peer review of "Fabrication and Characterization of Mineral Hydrophilic Antifogging Film via Vacuum Evaporation Method"

_coatings, doi:10.3390/coatings13040730_

Round 1
Reviewer 1 Report
Excellent findings are reported in the field of anti-fog coatings. The manuscript may be accepted addressing the comments mentioned in the attached manuscript.

Author Response
Dear reviewer:
Re: Manuscript ID: Coatings-2269180 and Title: Fabrication and characterization of mineral hydrophilic antifogging film via vacuum evaporation method
Thank you from the bottom of my heart for your recognition and comments on our manuscript. Those comments are valuable and helpful. We have read through comments carefully and have made corrections. Based on your valuable comments, we have uploaded the revised manuscript file. Revisions in the text are shown using red highlight for additions, and strikethrough font for deletions. The responses to the your comments are marked in red and presented following.
We highly appreciate your time and consideration.
Sincerely.
Shenyu Wei.
Point 1: Are these imaginations or being observed by AFM etc?
Response 1: Thanks for the reviewer’s comment. These are the assumed models based on the morphology observed by AFM. In order to make the model more clearly understood, I made some modifications to the line 263-267 in the text. These have been revised to “The theoretical values of the roughness factor formed by the cubic packing and the hexagonal closest packing model are discussed above. It can be seen from the results that the roughness factor is between 1.92 and 1.95 when the particles of different sizes are stacked. Similarly, we calculate the particle packing model with homogeneous size, whose roughness factor is between 1.51 and 1.81. Therefore, when particles of different sizes accumulate, the roughness factor is larger, and the surface of the material is rougher. Combined with the actual coating process, from the AFM results of Figure 5, it can be seen that when the film thickness is 100-200 nm, the sample surface has fluctuated mineral particles of different sizes, and the roughness of the film is relatively large. As the film layer gradually thickened to 800-1000 nm, the size and shape of the mineral particles deposited on the surface of the sample gradually became uniform, and the roughness of the film was small. Therefore, when the particles that make up the film are of different sizes, the surface of the material is rougher, so that the surface hydrophilicity is higher, which shows excellent anti-fog effect.”

Reviewer 2 Report
The part of the manuscript that described the preparation and characterization must be improved in order to become readable.
There are many typos that must be avoided.
As a conclusion the paper can be published after carefull rewriting the above mentioned part of manuscript.
Author Response
Dear reviewer:
Re: Manuscript ID: Coatings-2269180 and Title: Fabrication and characterization of mineral hydrophilic antifogging film via vacuum evaporation method
Thank you from the bottom of my heart for your recognition and comments on our manuscript. Those comments are valuable and helpful. We have read through comments carefully and have made corrections. Based on your valuable comments, we have uploaded the revised manuscript file. Revisions in the text are shown using red highlight for additions, and strikethrough font for deletions. The responses to the your comments are marked in red and presented following.
We highly appreciate your time and consideration.
Sincerely.
Shenyu Wei.
Point 1: The part of the manuscript that described the preparation and characterization must be improved in order to become readable.
Response 1: Thanks for the reviewer’s comment. The part of the manuscript that described the preparation has been revised to “The mineral hydrophilic anti-fog film is prepared as follows: 1) Experimental preparation: Prepare coating raw materials required for the experiment. The mineral powder was mixed with 5 wt% distilled water and put into a tabletting machine (YP-15, JOSVOK, Tianjin, China) at 40 Mpa for 10 min. Put the mineral billet into Muffle furnace (TMX-12-18, FNS, Beijing, China) for 2 h at 500°C, take out and cool to get the coating material. The polycarbonate (PC) substrate required for the experiment was ultrasonically cleaned with ethanol anhydrous (purity≥99.5 wt%, Macklin, Shanghai, China) and deionized water, respectively. The single ultrasonic cleaning time needed to reach 10 to 15 min, so that the oil and impurities on the surface of the substrate could be effectively removed. 2) Instrument preparation: The condensate circulation should be opened in advance before the vacuum coating mechanism (ZZS-900, Chengdu Vacuum Machinery Factory, Sichuan, China) is turned on. After opening the vacuum coating room, it is necessary to clean the coating room and keep it clean. Install the required coating coating material. After confirmation, close the coating room. 3) When the vacuum degree in the coating machine is 3×10-3-9×10-3 Pa, the coating operation begins. The substrate temperature is 60°C, and the coating rate is controlled to be 3-7 Å/s. The physical vapor deposition is used to vaporize the pressed raw material onto the polycarbonate (PC) base at high temperature, and the mineral material hydrophilic anti-fog film is obtained. Then the experimental operation is carried out according to the corresponding experimental parameters. After the entire coating process is completed, the sample can be taken out when the temperature in the coating room is lower than 50°C. After removal, close the coating room and keep the coating room. Finally, turn off the cold circulating water.” The part of the manuscript that described the characterization has been revised to “The TG-DTA8122 thermogravimetric analyzer (Rigaku, Kyoto, Japan) was used for differential thermogravimetric analysis of raw minerals. The test atmosphere was nitrogen atmosphere, the reference was blank, the heating rate was 10°C/min, and the scanning temperature range was 30°C-1000°C. The contact angle of the film surface to deionized water was measured by DSA100s contact angle tester (Kruss, Hamburg, Germany), which determine the hydrophilicity of the mineral hydrophilic anti-fog film. Deionized water was added to the syringe, and the drop amount was set to 4 μL / drop. Five different points on the coating were selected to test the contact angle, and the average value was calculated as the final contact angle of the coating. The surface morphology and surface roughness of the mineral hydrophilic antifogging film were observed by Dimension Icon atomic force microscope (Bruker, Karlsruhe, Germany). In the test, the scanning mode is tap mode, the silicon probe is used, the elastic coefficient is 1.7 N/m, and the scanning frequency is 1 Hz. The optical transmittance of mineral hydrophilic antifogging film and substrate was measured by UV-2600 ultraviolet-visible spectrophotometer (Shimadzu, Kyoto, Japan). The background is air, the scanning wavelength was 200-1200 nm.”
Point 2: There are many typos that must be avoided.
Response 2: Thanks for the reviewer’s comment. I have checked and corrected the typos in the text.

Reviewer 3 Report
Paper devoted to fabrication and characterization of mineral hydrophilic antifogging film via vacuum evaporation method. This topic interesting and relevant but there are some comments:
1. In the introduction it should be noted that hydrophilic films can also be obtained by electrochemical method and give links to the following publications:
- Tseluikin V.N. Electrodeposition and Properties of Composite Coatings Modified by Fullerene С60 // Protection of Metals and Physical Chemistry of Surfaces. – 2017. – V. 53, â„– 3. – P. 433 – 436.
- Lanzutti A., Lekka M., de Leitenburg C., Fedrizzi L. Effect of pulse current on wear behaviour of Ni matrix micro- and nano-SiC composite coatings at room and elevated temperature // Tribology International. 2019. V. 132, P. 50 – 61.
- Shourgeshty M., Aliofkhazraei M., Karimzadeh A. Study on functionally graded Zn–Ni–Al2O3 coatings fabricated by pulse electrodeposition // Surface Engineering. – 2019. – V. 35, â„– 2. – P. 167 – 176.
2. In my opinion, Materials and methods are described too briefly. It is necessary to describe it in more detail.
Author Response
Dear reviewer:
Re: Manuscript ID: Coatings-2269180 and Title: Fabrication and characterization of mineral hydrophilic antifogging film via vacuum evaporation method
Thank you from the bottom of my heart for your recognition and comments on our manuscript. Those comments are valuable and helpful. We have read through comments carefully and have made corrections. Based on your valuable comments, we have uploaded the revised manuscript file. Revisions in the text are shown using red highlight for additions, and strikethrough font for deletions. The responses to the your comments are marked in red and presented following.
We highly appreciate your time and consideration.
Sincerely.
Shenyu Wei.
Point 1: Paper devoted to fabrication and characterization of mineral hydrophilic antifogging film via vacuum evaporation method. This topic interesting and relevant but there are some comments:…
Response 1: Thanks for the reviewer’s comment. I have added the electrochemical deposition to the introduction. The text added is as follows. “Electrochemical deposition method is used to obtain materials with different morphologies and microscopic dimensions by adjusting parameters such as current density, deposition time, and temperature. It can adjust the superhydrophilicity of the material from structural and morphological directions. You et al. used electrodeposition method to achieve the preparation of Zn/ZnO crystals by electrodeposition on a copper grid. The electrodeposition process was optimized by varying the applied voltage and duration. The all-inorganic film exhibited superhydrophilicity in air and could be used as an efficient oil-water separation device.” (References are listed in the text)
Point 2: In my opinion, Materials and methods are described too briefly. It is necessary to describe it in more detail.
Response 2: Thanks for the reviewer’s comment. The part of the materials and methods has been revised to “The mineral hydrophilic anti-fog film is prepared as follows: 1) Experimental preparation: Prepare coating raw materials required for the experiment. The mineral powder was mixed with 5 wt% distilled water and put into a tabletting machine (YP-15, JOSVOK, Tianjin, China) at 40 Mpa for 10 min. Put the mineral billet into Muffle furnace (TMX-12-18, FNS, Beijing, China) for 2 h at 500°C, take out and cool to get the coating material. The polycarbonate (PC) substrate required for the experiment was ultrasonically cleaned with ethanol anhydrous (purity≥99.5 wt%, Macklin, Shanghai, China) and deionized water, respectively. The single ultrasonic cleaning time needed to reach 10 to 15 min, so that the oil and impurities on the surface of the substrate could be effectively removed. 2) Instrument preparation: The condensate circulation should be opened in advance before the vacuum coating mechanism (ZZS-900, Chengdu Vacuum Machinery Factory, Sichuan, China) is turned on. After opening the vacuum coating room, it is necessary to clean the coating room and keep it clean. Install the required coating material. After confirmation, close the coating room. 3) When the vacuum degree in the coating machine is 3×10-3-9×10-3 Pa, the coating operation begins. The substrate temperature is 60°C, and the coating rate is controlled to be 3-7 Å/s. The physical vapor deposition is used to vaporize the pressed raw material onto the polycarbonate (PC) base at high temperature, and the mineral material hydrophilic anti-fog film is obtained. Then the experimental operation is carried out according to the corresponding experimental parameters. After the entire coating process is completed, the sample can be taken out when the temperature in the coating room is lower than 50°C. After removal, close the coating room and keep the coating room. Finally, turn off the cold circulating water.”

Reviewer 4 Report
accepted in current form
Author Response
Dear reviewer:
Re: Manuscript ID: Coatings-2269180 and Title: Fabrication and characterization of mineral hydrophilic antifogging film via vacuum evaporation method
Thank you from the bottom of my heart for your recognition and comments on our manuscript. Those comments are valuable and helpful. We have read through comments carefully and have made corrections. Based on your valuable comments, we have uploaded the revised manuscript file. Revisions in the text are shown using red highlight for additions, and strikethrough font for deletions. The responses to the your comments are marked in red and presented following.
We highly appreciate your time and consideration.
Sincerely.
Shenyu Wei.